# Identification of the Mechanical Failure Factors with Potential Influencing Road Accidents in Ecuador

**DOI:** 10.3390/ijerph19137787

**Published:** 2022-06-24

**Authors:** Juan Pablo Montero-Salgado, Jose Muñoz-Sanz, Blanca Arenas-Ramírez, Cristina Alén-Cordero

**Affiliations:** 1Machine Engineering Division, Escuela Técnica Superior de Ingenieros Industriales (ETSII-UPM), Universidad Politécnica de Madrid (UPM), 28006 Madrid, Spain; joseluis.munozs@upm.es; 2Transportation Engineering Research Group, Universidad Politécnica Salesiana, Cuenca 010105, Ecuador; 3University Institute for Automobile Research Francisco Aparicio Izquierdo (INSIA-UPM), Universidad Politécnica de Madrid (UPM), 28006 Madrid, Spain; blanca.arenas@upm.es; 4Escuela Politécnica Superior, Universidad de Alcalá (UAH), 28871 Alcala de Henares, Spain; cristina.alen@uah.es

**Keywords:** Vehicle Technical Inspection, road safety, road crashes, mechanical failures, maintenance

## Abstract

Road traffic accidents result in injury or even death of passengers. One potential cause of these accidents is mechanical failures due to a lack of vehicle maintenance. In the quest to identify these mechanical failures, this paper aims to set up the procedure to identify the mechanical failures that contribute to traffic accidents in cities located in developing countries, including the city of Cuenca-Ecuador. For present research, a database provided by the entity responsible for the Vehicle Technical Inspection, the Empresa Pública Municipal de Movilidad, Tránsito y Transporte and for the ones responsible of managing traffic accident data, Oficina de Investigación de Accidentes de Tránsito and Sección de Investigación de Accidentes de Tránsito was used. The vehicle subcategories M1 and M3 (bus type) and N1, so named according to Ecuadorian technical standards, were considered the most relevant regarding accident rates. The database was analysed with descriptive statistics, a Pareto chart and time series with the quadratic trend. From this analysis, the most significant failures found in the VTI in all three subcategories were the alignment of the driver headlight, both horizontal and vertical, braking imbalance on the 2nd axle, insufficient tire tread and parking brake effectiveness. All these failures showed a decreasing trend over time and in the forecast at a maximum of two to three years. The most relevant causes of road accidents recorded during the period 2009–2018 related to mechanical failures were the braking system (65.5%) and the steering system (17.2%) for subcategory M1.

## 1. Introduction

One of the initiatives that succeed in reducing road crashes and injuries are the United Nations legal instruments developed with the support of United Nations Regional Commissions and used by many countries as a guide for the development and implementation of traffic regulations, the manufacturing of safer vehicles, the reduction of risk collision in vehicles transporting dangerous goods and, to ensure that only vehicles with the appropriate maintenance are on roads. In addition, the effort of the World Health Organization (WHO), which encourages nations to implement activities within the program known as “Safer traffic road users,” which suggests the activity of “*to establish and monitor the compliance of transport, health and work safety laws and the standards and rules for the safe operation of commercial freight and transport vehicles, road passenger transport services and the rest of the public and private vehicle fleet, to reduce the injuries caused by accidents*” [1], confirming the need of ensuring that only vehicles with the adequate maintenance may travel.

Ecuador, as well as other nations that implement WHO activities, has laws and standards that regulate Land Transport, Traffic and Road Safety, where it was established that “*Owners of motor vehicles are obliged to have them checked in technical and mechanical revisions at vehicle inspection and control centres, authorised in accordance with the regulations issued by the National Traffic Agency*” [2]. 

By Resolution No 006-CNC-2012, dated 26 April 2012, published in the Official Record Supplement No 712 dated 29 May 2012, the National Competence Council transferred the competence to plan, regulate and control traffic, land transport and road safety to the municipal decentralised autonomous governments of the country, providing them with all the competences over-involved processes, to improve mobility in their respective territorial areas.

The Vehicle Technical Inspection (VTI) was established in the city of Cuenca the 29 August 2005 by the Illustrious Cantonal Council of Cuenca through the “Ordinance that Rules the Establishment of the Vehicle Technical Inspection System of Cuenca and the Competence Delegation to CUENCAIRE”. The aim was to check the proper operation of vehicles, the level of pollutant gas emissions, the level of noise and the suitability when it was the case. Thus, a healthy environment is safeguarded at the same time as the ratio of accidents due to mechanical failure in vehicles [3]. CUENCAIRE Corporation implemented the VTI mandatory system in 2008 [4].

The Cantonal Council of Cuenca, on the 30 October 2012, issued the Ordinance where the Decentralised Autonomous Government of the city of Cuenca ratifies the delegation of traffic regulation and control, transport and road safety to the Empresa Pública Municipal de Movilidad, Tránsito y Transporte (EMOV EP). Cuenca has two VTI centres (Mayancela and Capulispamba), operated by the Danton Consortium, where citizens, prior to the vehicle registration, must attend and approve the VTI [5]. This inspection establishes several control stages, such as visual inspection and the measurement of certain mechatronic factors. Moreover, vehicle pollutant emissions are controlled.

According to the above, the need to assure that only properly maintained vehicles can circulate is confirmed. Thus, in Ecuador and, therefore, in the city of Cuenca, it is mandatory that the owners of motor vehicles are obliged to carry out the technical vehicle inspection, with the aim of reducing the proportion of traffic accidents due to this cause.

The high percentage of deceased or victims with injuries caused by traffic accidents has enormous consequences for communities all over the world. Consequently, this issue requires to be analysed at worldwide, national and local levels.

The problem is getting worse worldwide. The number of deaths caused by road crashes reached 1.35 million in 2016, which implies that 3700 people die on the world’s roads every day. Injuries caused by road crashes are the eighth leading cause of death for all age groups and it is currently the leading cause of death for children and young people aged 5–29 years. World leaders committed to halving the number of deaths due to road crashes by 2020. But little progress has been made towards this goal. Therefore, there is a need to scale up evidence-based interventions and investments [6].

Most recent available data (2016) in the Region of the Americas, where the city of Cuenca-Ecuador is located, show that death caused by traffic accidents was the second leading cause of mortality in young adults aged 15–29 years. Injuries by road traffic caused the death of around 154,997 people, meaning 11% of the worldwide deaths. It should be highlighted that between 2013 and 2016, in four middle-income countries (Argentina, Ecuador, Paraguay and Venezuela), there was no change recorded in the deaths caused by road traffic, with Ecuador’s mortality rate caused by road traffic as the eighth highest in the Region of the Americas, being far above the regional mortality rate, that is of 15.6 per 100,000 inhabitants [7].

Road transport is the main and key means for the development of the countries, enabling the mobility of the majority of people. Private vehicles and public transport are the means with the highest share of road mobility. Private vehicles have the highest representation in the countries’ vehicle fleet and as vehicles involved in road crashes. The risk of dying in a road crash as private vehicle or bus passengers is 10 times higher [8]. According to the Ecuadorian technical standard NTE INEN 2656, private vehicles are classified as subcategories M1 and N1 and buses as subcategory M3 bus type (ISO 3833 terms and definitions). According to the statistics provided in [9], in 2018, 70% of the vehicle fleet in Ecuador was formed by M1 (47.70%), N1 (20.68%) and M3 bus types (1.16%).

On the other hand, a total of 25,530 road crashes were recorded in Ecuador in 2018, with 19,850 injured and 2151 deceased, being 74% of them the driver of the vehicle. It should be noted that 59% of the accidents corresponded to private cars and buses. That same year, in the province of Azuay, where the city of Cuenca is placed, the number of road crashes was 1528, with a total of 1357 injured and 102 deceased. In 24.13% of accidents, the probable cause recorded, named as driving inattentive to traffic conditions (mobile phone, video screens, food, make-up or any other distracting factor), meanwhile the one known as mechanical failure in the systems and/or tyres (braking system, electronic or mechanical steering) was 0.82% [10]. Unfortunately, there is no available information on accident causes neither by province nor by vehicle type.

The authors [11] conclude that in the period 2014–2019, there was a 31.12% decrease in the number of road crashes in Ecuador, while the number of deaths increased by 1.8%. These data distance Ecuador from achieving the Sustainable Development Goals 2020 target of 3.6 and require adopting measures for the prevention of road crashes and their consequences to comply with the new decade regarding road safety until 2030.

Most regions and countries in the world have “agreed” to work together to achieve the goal of reducing road deaths and one of the pillars is safe vehicles. In countries of the Region of Americas, fleet renewal, as well as the maintenance of vehicles, has to be observed by the administration as one of the most important factors for safety.

Attending the percentage of traffic accidents related to the cause named as *mechanical failure in systems and/or tyres (braking system, electronic or mechanical steering)*, regarding the percentage of injures (average of 0.53%) and deceases (average of 0.65%), for the latest available data (January 2019–December 2019) from the Agencia Nacional de Tránsito del Ecuador (ANT), it can be observed that, although the percentages are not high, they remain over time with a remnant value, except for the month of August where no data is recorded, possibly due to a management problem. In addition to this, according to [9], from 2013 to 2018, the vehicle fleet showed an average annual growth in the subcategories M1 of 6.66%, M3 bus type of 24.54% and N1 of 6.77%.

Vehicle safety inspections have not been widely studied as a tool, even though they can help both to reduce fatal crashes and keep roads safe, as they require vehicle owners to keep their vehicles in good operating condition [12].

Under the statement “Talking about safety in the Mechanics and Machinery industrial sector not only means health and safety at work but also maintaining safety in domestic and leisure activities” [13], some questions are raised and answered, being one of them “What are the most frequent failure modes and non-compliances that can be found on the current reality?”. Under this perspective, the objective of this research is to set up the procedure to identify the mechanical failures that contribute to traffic accidents and try to give an answer to this question in cities located in developing countries, in the context of the vehicle fleet in the city of Cuenca-Ecuador. This requirement proposes to establish an analysis of the information provided by the entities responsible for the VTI and those in charge of managing road crash information in the city of Cuenca-Ecuador, in addition to the fact that the latter do not have organised data, so there is a work of collecting information from paper files and digital files.

In this article, an analysis of the data recorded in the VTI Centers and road crashes of M1 (vehicles), M3 (buses) and N1 (pickups and vans) vehicles subcategories is done because they are over-represented in road crashes rates in 2018, according to the ANT.

Noting the information provided by the entities responsible for the VTI and those in charge of managing road accident information will enable this paper to provide an insight into the most typical mechanical failures that occur in accidents or that could cause them and, in the future, to establish possible links between the importance of vehicle maintenance, the prevention of road crashes and the improvement of road safety.

## 2. Literature Review

In [12], vehicle registration data of Pennsylvania in the United States are combined with two large samples of state safety inspection results, which monitor the safety of vehicle components because, as with any part subjected to stress or wear, maintenance is necessary for proper operation. Vehicle safety in terms of maintenance is identified by analysing the failure rate trends over time. It is concluded that, despite the fact that technological improvements introduced in recent years make vehicles appear safer, the failure rate found on vehicles in VTI does not trend to zero in the near future.

Many countries see the Vehicle Technical Inspection as a possible control tool to decrease traffic accidents caused by mechanical failures. However, ref. [14] obtained results that indicate that the VTI does not have a significant impact on the technical condition of the vehicles with a certain age of operation nor in a significant increase in the interventions performed in the vehicle repair industry, suggesting that the inspection does not improve the mechanical condition of vehicles. In this study, the results suggest that periodic vehicle inspection is a poor instrument to achieve the objectives set out at the political and social levels. In addition, ref. [15] concludes that vehicle safety inspections do not represent an efficient use of the funds and do not seem to have a significant mitigating effect on the role of vehicle failures in road crashes. Despite this, VTI should not be ruled out as a solution, as sector conditions and the legislation regulating vehicle traffic in each country are different.

The authors [16] concluded that there is a relationship between the vehicle’s technical condition and road crash risk. This work underlines the need to ensure a certain level of reliability in terms of vehicle technical condition through operation and maintenance systems and a stricter regulation as many countries already do, including environmental safety monitoring, by requiring good vehicle fleet condition with appropriate maintenance.

In study [17], the relationship between vehicle mechanical failures and road crashes in developing countries is addressed. They conclude that the use of non-original parts and poor quality supervision during and after repairs could cause road crashes. This work highlights the importance of reliability and monitoring of the correct vehicle maintenance to increase road safety in some countries.

The hypothesis that vehicle fleet good condition decreases road crashes is also supported by the work of [18], which underlines the important role of periodic vehicle inspection in the future probability of collisions. It also recommends that authorised heavy vehicle inspection centres provide driving safety instructions, training and even specific evaluations to their drivers.

Studies in developed and developing countries have shown that one of the main causes of road crashes are mechanical failures in vehicles [19,20,21], therefore accidents that could be prevented by the extension and implementation of VTIs, with mandatory regulation where it does not yet exist.

In Cameroon, one of the main causes of fatal collisions was mechanical failure (28%), two-thirds of which were tyre problems [22]. It is worth highlighting that this cause of crashes can be prevented through appropriate interventions by the entities in charge of controlling the attendance of vehicles in VTI centres and thus contributing to the improvement of road safety.

In [23], implement statistical significance tests to investigate the effectiveness of VTIs on road safety matters in the United States. The results indicate that the States with inspections record a lower number of monthly vehicle crash claims than the States without inspections, which indicates a positive effect on road safety.

In the study by [24], a systematic review was performed to determine the effect of VTIs on road crashes and injuries and experiences with fewer or non-periodic inspections were compared. The authors suggest that periodic inspection is associated with a slight decrease in road crashes and they outline the need to conduct further studies that provide conclusive scientific evidence.

Traffic accident analysis in a community aims to be able to implement measures to decrease its consequences. Ref. [25] introduce a systematic review and meta-analysis of studies estimating the relationship between the number of accidents involving motorised vehicles and cyclists or pedestrians and the motorised vehicle, cyclist and pedestrian traffic volume, demonstrating the efforts to determine different interactions between variables that could be the cause of traffic accidents.

As aforementioned, means of transport are of great importance, as they have contributed to the economic development of countries and have made the movement of goods and people possible, enabling social integration. Therefore, studies about accidents in means of transport have been developed, such as the one by [26], who contemplates that safety can be defined as an acceptable state of risk taken by society.

According to the data provided by the institutions in charge of road safety, sometimes these are not properly organised, or in other cases, they are scattered in paper and digital files, which is why [27] states that “the methodology of data preparation from the records of the VTI centres, allows establishing a suitable database for its use in mobility analysis”. Access to reliable and systematic data sources is an unavoidable condition to move forward in the knowledge of mobility, fleet condition and road safety in regions and countries such as Ecuador.

Moreover, the use of statistical tools on a structured, systematic and high-quality database allows the analysis of the behaviour of the variables of interest, such as the one carried out by [28,29], which uses variables such as age, gender, type of road crashes and driver offences, including vehicle compliance with VTI. An understanding and analysis of the factors involved in road crashes can help road policy decision-making and the development of preventive measures, as well as raising drivers’ awareness of the need to comply with the VTI.

Considering the study of [30], older vehicles and vehicles with defects represent a not negligible volume of vehicles running in deficient conditions, which deserves the attention of political decision-makers. The results obtained support the need to establish control tools to limit the operation of these vehicles.

In [27], a sample of the data recorded in the VTIs is used and it is established that in the case of private cars, vehicle age is the most important factor in mobility.

On this premise, ref. [31] provides data regarding the average age in years of the vehicle fleet in Ecuador from 2012 to October 2021, showing an increase in the age of the vehicle fleet from 13.26 years in 2012 to 16.2 years in 2018. Although there seems to be some vehicle turnover from 2018 onwards, as in 2021, the average age is 15.9, this change is very short. According to [32], insurance companies in Ecuador consider that a car older than 12 years is already old, and, therefore, they do not insure it, or it is charged with higher premiums. Considering the average age of 15.9 years and the fact that these vehicles still operate on the roads, the age of the vehicle fleet in Ecuador is a factor that highlights the need for strict and sustained control of the performance of the VTIs.

## 3. Materials and Methods

### 3.1. Data Collection for the Research

The official source of statistical information in Ecuador is the Instituto Nacional de Estadística y Censos (INEC), which publishes its results annually, but the data are unreliable [11]. For this reason, the authors turned to the entities in charge of Vehicle Technical Inspection and governmental entities in charge of investigating traffic accidents in the city of Cuenca, with which the academy has agreements to consolidate the relevant information for this research.

The data regarding Vehicle Technical Inspection (VTI) records were provided by the EMOV EP. The information corresponds to the period 2008–2018. It includes all items considered in the VTI process, omitting data such as vehicle plate number or chassis number for data protection. Figure 1 shows the process used for data collection. However, it is noted that the entity does not have up-to-date data due to the COVID 19 pandemic and the slow vaccination process in Ecuador.

Road crash data required for this paper were provided by government entities in charge of researches of this type of accidents in the city of Cuenca-Ecuador, which are the Oficina de Investigación de Accidentes de Tránsito (OIAT) and the Sección de Investigación de Accidentes de Tránsito (SIAT). Two databases were obtained. One by means of photographic support of the technical expert reports recorded in the entities and the other from a digital file with the information of all the Azuay province, where the city of Cuenca is placed. Both databases correspond to the period 2009–2018 and the process used to obtain the data is represented in Figure 2. Because of the pandemic situation, there are no updated data available and some issues will have to be addressed as the data homogenisation because of the so peculiar characteristics of the year 2020.

### 3.2. Database Creation Required for the Research

For the database creation regarding mechanical and electrical failures of the VTI records, the first step is to discriminate the data that is not of interest, such as gas emission values, noise level values, chassis condition observations, bodywork, paint, plates, safety accessories, glass, vehicle interior and exhaust system parts.

Then, the categories of vehicles that are out of the scope of the research are discriminated and the remaining vehicles are given the corresponding subcategories: M1, M3 and N1.

From the remaining information, only the values that show the defect rating are selected, focusing on those which do not fulfil the rating thresholds based on the specific standards that regulate the component to be evaluated, used by the EMOV EP in the VTI. This process has been repeated year to year of VTI since 2008 until obtaining the database under study. This process can be observed in Figure 3.

The methodology used to generate the traffic accident database is based on the collection of information through field research. In its subsequent processing, the information is classified and segregated, obtaining the required database. In the case of the physical files, first of all, photographs of the existing technical expert reports existing at the OIAT and SIAT facilities are taken. The photographs are then classified according to the subcategories of the vehicles of interest for the research. Subsequently, the variables required for the creation of the database are chosen. Finally, the necessary information is extracted from the photographs related to these variables. In the case of digital data files, the first step is to propose the necessary variables for the database that will be analysed later. These variables are the same raised in the analysis of the physical files. Then, the sites or places that do not belong to the city of Cuenca are discriminated against. Subsequently, data from the vehicles belonging to the subcategories of interest is extracted for the study. Finally, the traffic accident database is obtained. The process is shown in Figure 4.

### 3.3. Statistical Procedures

Once the OIAT, SIAT and EMOV EP databases are structured, the statistical analysis is carried out to establish the most relevant mechanical and electrical defects in the VTI records and the causes of traffic accidents. Several statistical procedures are implemented.

#### 3.3.1. Descriptive Statistics

Descriptive statistics is a fundamental part of data analysis and provides the basis for the comparison, being part of the good practices for the collection of relevant information and for the process of researching causes and influencing factors. An overview of the corresponding data is obtained, either by means of the mode (the value or category of the variable that is most often repeated or has a higher frequency) or by histograms (graphical representation of a frequency table indicating the number of cases occurring in each value of the variable) [33].

The implementation of descriptive statistics and its tools is used in the transport field, for example, in the paper by [34] to explore the effect of public transport strikes on traffic conditions, to study the impacts of average speed and travel time on traffic flow, etc. The study by [35] examines the health-related behaviours of e-scooter users, using descriptive statistics of the variables (age, gender and area of residence) and chi-square analysis to determine the relationships between the variables and the equivalence of proportions. Furthermore, ref. [36] implement descriptive statistics to determine the characteristics of road crashes with autonomous vehicles in terms of the accident location, weather conditions, way of driving, vehicle movement before the road crash occurred, speed, type of collision, crash severity and location of vehicle damage.

In this research, the analysis allows one to determine which are the most frequently occurring mechanical and electrical defects in the VTI records and the histogram allows one to visualise the frequency of the different causes of accidents recorded for each type of vehicle selected.

#### 3.3.2. Pareto Chart

The Pareto chart is an important tool given its capacity to help to focus attention on the area(s) of interest. It is useful for separating the few significant factors from the many trivial ones, identifying the most important sources of problems helping to prioritise and assign resources [37].

In the transport area, the Pareto chart is used by the authors [38] to establish the most influential factors in the analysis of energy absorbers considered as a solution to minimise the consequences of road crashes on passengers and improve automobile safety. In the paper of [39], the Pareto chart is used for the microscopic simulation of traffic to investigate traffic efficiency and road safety. Authors in [40] implemented the Pareto chart to identify the causes of failure modes in the tyre braking system in vehicles of a commercial vehicle company.

In this research, the Pareto chart enables identifying the most relevant failures of the vehicles in selected subcategories which cover 80% of the total failures found in the VTI process and the mechanical failures which cause traffic accidents. Since the database of the VTI uses very long names for each failure, an alphanumeric coding for the Pareto chart was proposed. The full table is included in Appendix A.

#### 3.3.3. Time Series

Time series analysis deals with statistical methods for analysing and modelling an ordered sequence of observations, which is usually over time (equally spaced time intervals). This allows the understanding and modelling of the data generation system and the forecast of future values. By means of the graphic representation of data in chronological order, it is possible to determine if there is a trend or a pattern, adjusting data to a certain curve type [41].

Time series are useful for analysing several issues in the transport area, as in the paper by [42], where the characteristics of fatal road crashes in a developing country are explored and accident predictions are also obtained. In [43], time series is used to explain and predict the relationship between the temporary variation of Safety Performance Functions (SPFs). SPFs are widely used by state and local authorities to predict crashes. In [44], a method of speed prediction of the moving vehicle at high speed with autoregressive integrated moving average (ARIMA) models is proposed.

This work uses time series analysis with the quadratic trend, this being the one that best fits with the most representative failures recorded in the VTI. It should be highlighted that the R2 metric was used to determine the proper fitting of each of the established models. This allowed us to determine and predict the behaviour of each of the failures over time.

## 4. Results

### 4.1. Mechanical and Electrical Conditions of Vehicles for Subcategories M1, M3 and N1, of the City of Cuenca-Ecuador from 2008 to 2018

VTI data for the period 2008–2018 obtained in the research by applying the mode in each of the subcategories N1, M1 and M3 bus type indicate as a result that the most significant defects are: horizontal alignment of the driver headlight, vertical alignment of the driver headlight, braking imbalance on the 2nd axle, insufficient tyre tread and parking brake effectiveness, results that are shown in Table 1. These results are the key parameters for the models discussed in Section 4.2.

By implementing the Pareto chart analysis to each subcategory, it is obtained that, for M1 vehicles (Figure 5), the most representative failure is the vertical alignment of the driver headlight (19.4%), followed by horizontal alignment of the driver headlight (17.4%). Both failures were located on the driver headlight with a cumulative percentage of 36.8%. It is also observed that 11.7% of the vehicles fail in parking brake effectiveness and 8.5% have insufficient tyre tread. Likewise, and in decreasing order, it is obtained that 8.1% fail in braking effectiveness, 7.1% have braking imbalance on the 2nd axle, 4.1% fail in the alignment of the 1st toe-in axle, and, finally, 3.1% of the failures are due to braking imbalance on the 1st axle.

For vehicles of subcategory N1, Figure 6 shows that the failure with the highest percentage value is the vertical alignment of the driver headlight (13.5%), followed by the horizontal alignment of the driver headlight (11.8%). Both failures located in the driver headlight give a cumulative percentage of 25.3%. There is also parking brake effectiveness (9.9%), insufficient tyre tread (7.2%), left wheel suspension effectiveness on the 1st axle (6.3%), braking imbalance on the 2nd axle (5.5%), right wheel suspension effectiveness on the 1st axle, inadequate fitting of spring eye bushings and braking effectiveness (around 4% each), and suspension imbalance on the 1st axle, braking imbalance on the 1st axle and suspension imbalance on the 2nd axle (all of them with 3%), being the remaining failures around 2%.

Figure 7 shows the results for the subcategory M3 bus type. For this subcategory, the most representative failures are suspension effectiveness on the right wheel of the 1st axle, suspension effectiveness on the right wheel of the 2nd axle, suspension effectiveness on the left wheel of the 1st axle, suspension effectiveness on the left wheel of the 2nd axle, each one with a percentage of around 10.5%. Braking effectiveness (7.7%), vertical alignment of the driver headlight (6.5%), horizontal alignment of the driver headlight (4.9%), braking imbalance on the 2nd axle, inadequate fitting of spring eye bushings and braking imbalance on the 1st axle (around 4%). Finally, the following with almost 3% are: sets or wear in the steering bars and in the steering box.

Establishing in a clear way the most frequent failures and non-conformities that can be found when performing a Vehicle Technical Inspection provides as a consequence security to the citizens in general regarding vehicle traffic, in accordance to what [45] indicates in the statement “*This involves verifying compliance with the essential health and safety requirements related to the design and construction of machinery and, to test the machine accordingly to the related harmonised standards*”.

### 4.2. The Most Representative Mechanical and Electrical Failure Trend for Subcategories M1, M3 and N1 Vehicles in the City of Cuenca, Ecuador from 2008 to 2018

Once set up the most representative failures within the different subcategories, an analysis of the most common ones is performed, with the aim of extrapolating vehicle conditions from the subcategories under study.

For the development of this analysis, the time series method with quadratic trend was used, forecasting the behaviour of the most recurrent failures in the database for each of the years studied, being the horizontal alignment of the driver headlight, vertical alignment of the driver headlight, braking imbalance on the 2nd axle, insufficient tyre tread and parking brake effectiveness. It should be noted that a good fit of each of the models to the data was obtained with an R2 above 0.8, generating reliable forecasts.

The time series forecast with the quadratic trend for this analysis was performed at a maximum of either two or three years, as the projections showed negative values.

#### 4.2.1. Horizontal Alignment of the Driver Headlight Failure Trend

Figure 8a shows the behaviour of the failure named as *horizontal alignment of the driver headlight (AHFC08)* in the period 2008–2018 (blue line). From the set of values, there are three visible data (red dots) that are outliers due to the fact that in the years 2008 and 2009, there was low vehicle attendance VTI. Particularly in 2011, the Ecuadorian Technical Standard 1155 was applied, which established the requirements for devices to maintain or improve visibility in motor vehicles, to review more strictly the alignment of the headlights in the ITV in the city of Cuenca. This standard came into force nationally in 2012, so users to avoid penalties before this new regulation performed preventive maintenance to the headlights of the vehicle. For these reasons, these values have been discriminated for the curve fitting determined in Figure 8b by the time series with a quadratic trend (red line) and the three-year forecast (red dots).

In 2018, there were 39,904 vehicles with this defect, but it is estimated that by 2021 there will be 25,168.73 vehicles with this defect, that is, a percentage decrease of 36.92%.

#### 4.2.2. Vertical Alignment of the Driver Headlight Failure Trend

Figure 9a shows the behaviour of the failure named *vertical alignment of the driver headlight (AVFC09)* in the period 2008–2018 (blue line). Again, the same three outliers (red dots), which were already seen in Section 4.2.1, are visible. As the two types of failure are related to the same part (headlight), they have not been considered for the curve fitting determined in Figure 9b by the time series with a quadratic trend (red line) and the three-year forecast (red dots).

In 2018, there were 43,777 vehicles with this defect, but it was estimated that by 2021 there will have been 19,972.58 vehicles with this defect, that is, a percentage decrease of 54.37%.

#### 4.2.3. Braking Imbalance on the 2nd Axle Failure Trend

Figure 10a shows the behaviour corresponding to the failure named as a *braking imbalance on the 2nd axle (DF2E03)* in the period 2008–2018 (blue line). From the set of values obtained, there are two outliers (red dots) due to the low number of vehicles performing the VTI in the years 2008 and 2009. These data are not considered for the curve fitting determined in Figure 10b by the time series with a quadratic trend (red line) and the three-year forecast (red dots).

It is shown that in 2018, there were 15,143 vehicles with this failure, but it is estimated that by 2021 there will be 5655 vehicles with this defect, that is, a percentage decrease of 62.65%.

#### 4.2.4. Insufficient Tyre Tread Failure Trend

Figure 11a represents the behaviour corresponding to the failure named as *insufficient tyre tread (ILN02)* in the period 2008–2018 (blue line), while Figure 11b shows the fitting curve determined by the time series with a quadratic trend (red line) and the two-year forecast (red dots).

It is observed that in 2018 there were 14,451 vehicles with this defect. And it is estimated that by 2020 the number of vehicles with this defect would be 4785, that is, a percentage decrease of 66.88%.

#### 4.2.5. Parking Brake Effectiveness Failure Trend

Figure 12a shows the behaviour corresponding to the failure named as *parking brake effectiveness (EFE02)* in the period 2008–2018 (blue line), while Figure 12b shows the fitting curve determined by the time series with a quadratic trend (red line) and the two-year forecast (red dots).

It is noted that in 2018, there were 14,594 vehicles with this failure. It is estimated that by 2020, the number of vehicles with this defect would be 421, that is, a decreased percentage of 97.11%.

### 4.3. Vehicle Traffic Accidents for Subcategories M1, M3 and N1, of the City of Cuenca, Ecuador from 2009 to 2018

During the research, the number of accidents per year in the city of Cuenca of each of the subcategories under study was obtained in the offices of the SIAT and OIAT entities. In this compilation, there is a ceding of the management responsibilities in the handling of information between both entities in 2013. Figure 13 shows the number of accidents for subcategory M1, highlighting 2010 as the year with the highest number of accidents recorded by the SIAT entity, while 2013 showed the highest number of accidents recorded by the OIAT entity. In Figure 14, it can be observed that there is no record of traffic accidents for 2009 in the subcategory M3 bus type. Moreover, it can be seen that 2010 is the year with the highest number of accidents recorded by the SIAT entity, while for the OIAT entity, that year was 2013. Finally, Figure 15 focuses on subcategory N1. It can be seen that there is no record of traffic accidents for 2009, highlighting 2010 as the year with the highest number of accidents recorded by the SIAT entity, being 2015 for the OIAT. In all three cases, there is a decreasing trend in the number of records until 2018. Traffic accidents prior to 2009 could not be collected because the entities did not have records before this year.

### 4.4. Traffic Accident Causes of Vehicles for Subcategories M1, M3 and N1, of the City of Cuenca-Ecuador from 2009 to 2018

The research enabled us to establish the causes of traffic accidents in each of the subcategories under study. Figure 16, Figure 17 and Figure 18 shows the number of accidents and their causes for subcategories M1, M3 bus type and N1, respectively.

It can be noticed that the most representative cause of accidents in the three subcategories is a distraction at the steering wheel (driving inattentive to traffic conditions due to the use of mobile phones, video screens, food, make-up or any other distracting factor). While the cause described as a mechanical failure in the systems and/or tyres (braking system, electronic or mechanical steering) appears in six of the ten years analysed for subcategory M1, for the M3 bus type, there was only one accident in 2015. For N1 one, this cause appears only in three of the ten years analysed.

Considering data behaviour, it is worth mentioning what [46] says: “The fact that vehicle defects contribute to the occurrence of a crash, in some cases it is undeniable” and, as stated by [47]: “the proportion of crashes in which vehicle defects play a role is also not easy to estimate and will be underestimated by a significant degree in official crash statistics as police attending a crash do not normally have the time, training or motivation to examine a vehicle in-depth.” The stated fits with what was found in Ecuador at the time of an accident, so there is the possibility that the cause typified as driving inattentive to traffic conditions in some road crashes may not be the correct one.

When analysing the Pareto chart, the 29 accidents registered in 2009–2018 for subcategory M1 caused by mechanical failure, going in deep into the official accident report, it can be observed that the most relevant causes are braking system failure (65.5%) and steering system failure (17.3%), as shown in Figure 19. For subcategory M3 bus type, only one accident was registered due to mechanical gearbox failure in 2015. While for subcategory N1, which has four accidents due to mechanical failure (one in 2010, two in 2013 and one more in 2015), the causes described in the official report were respectively, electrical system failure (2010), braking system failure (both in 2013) and driving a vehicle in poor condition (in 2015), this last conclusion being highly vague.

## 5. Discussion

In each of the time series with quadratic trend analyses performed, a decrease in the occurrence of such failure is shown for future screenings. What is obtained in these analyses is related to technical vehicle inspections, as [46] states: “Periodic inspection regimes can be seen as a way of fleet quality control. An important question with any quality control system is the ability of the system to detect and repair failures reliably”.

This decrease may answer the question stated at the beginning of this research regarding the ability of the revision system to detect and subsequently proceed to correct the failure. Adding to this is the importance of appropriate maintenance to prevent failures that could even be the cause of a traffic accident.

When the situation in Norway was evaluated, ref. [48] established that a considerable decrease in the number of failures detected was obtained, indicating that the inspection schedule, together with the repairs associated with such inspection, decreased the occurrence of vehicle failures, something that is also seen in the trend analyses performed for the city of Cuenca-Ecuador.

Ref. [23] recommends that: “The states should maintain a vehicle safety inspection programme to reduce crash outcomes from the number of vehicles with existing or potential conditions”. Despite the fact that the above research regarding the effect of safety inspection programmes on crash outcomes is inconclusive, this recommendation, together with the results obtained in the current study, confirm that it should be adopted for Ecuador.

Moreover, it is added that technical vehicle inspections can identify unsafe vehicles and enable vehicle owners to perform repairs or remove these from traffic use. Without mandatory inspections, unsafe vehicles could continue running freely on the road [49], reaffirming the need for VTI in the city of Cuenca and in the country.

However, vehicle user behaviour is crucial regarding its maintenance, as it must fulfil certain minimum maintenance standards to be able to run on the roads. Therefore, it is essential to encourage citizen participation in VTI by means of the law and with its respective sanctions. In this context, ref. [28] conclude by pointing out that: “Vehicle mechanical conditions can play a determining role in road driving safety, which is why periodic Vehicle Technical Inspections are mandatory in many countries. However, the high number of drivers penalised for non-compliance with these standards is surprisingly high and there is not much evidence about what type(s) of reason(s) can explain this worrying scenario”.

Ref. [50] states that the technical condition of all vehicles deteriorates over time in one way or another. Moreover, it sets up that the evaluation results scope of vehicle technical suitability for road traffic in certain conditions depends mainly on the country-specific characteristics, approach and standards establishment in each region. It also mentions as additional influential parameters the specific conditions of the location where the VTI is implemented and the age of the vehicle, as has been considered in the case of the city of Cuenca-Ecuador, based on average age information provided by [31].

As shown in Section 4.2, the mechanical failures detected in the Vehicle Technical Inspection (VTI), which can be the cause of a traffic accident, have a decreasing trend in all analysed subcategories. However, taking into account the average age of the vehicle fleet running in Ecuador, it would seem hardly recommendable to eliminate the VTI performance requirement if the objective given by [50] is to be achieved, eliminating the possibility of having vehicles that are technically unroadworthy, which could represent a risk for the users.

Based on the present research, further studies should be carried out in the future to establish a clear correlation between VTI and traffic accidents, taking into account other soft computing techniques, as in [51], that use neural networks to predict the risk of collision on expressways or as in [52,53] that use these techniques to determine the effect of certain parameters of interest on the impact on the environment, being a powerful methodology and applicable in different areas.

## 6. Conclusions

Besides the Vehicle Technical Inspection research, the analysis performed regarding the data from the period 2008–2018 shows that the most common failures occurring in all vehicle subcategories are: horizontal and vertical alignment of the driver headlight (30%), braking imbalance on the 2nd axle (10%), insufficient tyre tread (6%) and parking brake effectiveness (7%), indicating the possibility of traffic accident occurrence if the user does not take the appropriate maintenance measures.

According to the analysis using the Pareto chart, subcategories M1 and N1 coincide in the fact that the failure with the highest percentage value is driver headlight alignment, being 36.8% for subcategory M1 and 25.3% for subcategory N1. Both subcategories have similar characteristics regarding the use, weight and size, being obtained in many cases by the same type of user. Considering this circumstance, the high rate of this failure is due to the fact that in Ecuador, being a developing country, users carry out headlight alignment in the automotive electrical service centres instead of taking it to a technical service of the vehicle brand to lower maintenance costs. These centres usually do not have the appropriate instrumentation and the calibration of measuring equipment is carried out empirically by persons who are not familiar with the head-light fitting standards.

In the case of subcategory M3 bus type, according to the Pareto chart analysis, the most representative failure is the effectiveness of the suspension on the four wheels of the vehicle, each one of them with a percentage of 10.5%. This failure related to the vehicle suspension system wear is due to the type of roads where they usually run. Most recent available data (2019) establishes that the condition of the road surface of the provincial road network of Azuay, where the city of Cuenca is located, is mostly in fair condition with 3394.04 km (67.14%), followed by good condition with 934.48 km (18.56%) and finally poor condition with 722.98 km (14.30%) [54]. The historic centre of the city of Cuenca has roads with irregular surfaces, cobblestones and roads around the city, considered of the second category (ballast or land roads), which do not have constant maintenance of the road surface, resulting in an irregular surface.

As an evident result of the time series analysis with the quadratic trend, it is obtained that each of the most representative failures in the three subcategories under study tends to decrease in the short term. Therefore, by 2021, both the horizontal and vertical alignment of the driver headlamp will decrease by 36.92 and 54.37%, respectively. Both failures belong to the same part (the driver headlight) and have a trend curve with the same behaviour, having disregarded the data for years 2008, 2009 and 2011 due to the reasons explained, such as the low number of vehicles in the VTI or the existence of errors in the data collection by the operator of the measurement equipment, possible failure in the measurement equipment or even a traceability problem in the equipment calibration. In the case of failure due to braking imbalance on the 2nd axle, a decrease of 62.65% is estimated for 2021, having disregarded the data for 2008 and 2009 due to the low number of vehicles attending the VTI. The three failures mentioned show identical behaviour in their trend curves.

However, for failures due to insufficient tyre tread and parking brake effectiveness, the quadratic trend model indicates that they will have decreased in 2020 by 66.88% and 97.11%, respectively. In both cases, negative values are obtained when calculating an estimate for the year 2021 as the fitting curve shows a very accentuated decreasing gradient. This fact is due to the fact that checking the condition of the tyres as well as the effectiveness of the parking brake are defects that are easily checked and repaired by the user of the vehicle before the VTI to avoid a possible sanction for not complying with the approval requirements. So that this defect is not seen in the VTI, leading to the conclusion about the effectiveness of the VTI in the detection and reduction of this type of failure.

The number of traffic accidents that occurred in the city of Cuenca was obtained in the research through physical and digital files placed in the offices of the SIAT and OIAT entities. In this collection, there was a transition of competences in the information management between both entities in 2013, which may have caused a certain loss of data. However, the information collected establishes a model of traffic accident behaviour in the three subcategories M1, M3 bus type and N1, showing in all cases a decreasing trend in the number of records in the analysed period 2009–2018.

As the result of the accident research, it was established that the most representative cause in the three analysed subcategories is a distraction at the steering wheel (due to the use of mobile phones, video screens, food, make-up or any other distracting factor). While for subcategory M1, the traffic accidents resulting from the cause described as mechanical failure in the systems and/or tyres (braking system, electronic or mechanical steering) come out in six of the ten years analysed, in subcategory M3 bus type there is only one accident in 2015. In subcategory N1, it happens in three of the ten years analysed. However, considering the circumstances of the accident, ref. [48] suggests that an influencing factor may be that a certain driver has a low-risk estimation together with high tolerance to vehicle failures, or if he/she can compensate for the known vehicle failures by driving in a conservative way. As a consequence, behavioural and attitude factors of the driver may mislead crash risk estimates associated with vehicle failures.

In the analysis of accidents caused by mechanical failure in the period 2009–2018, attending to the cause indicated in the official accident report, it could be concluded that the most repeated causes for subcategory M1 are braking system failure (65.5%) and steering system failure (17.2%). For the two remaining subcategories (M3 bus type and N1), there were few accidents due to mechanical failures making it difficult to obtain any relevant conclusion. This fact is due to the fact that in Ecuador, being a developing country, the drivers involved in this type of accident reach agreements to avoid the detention of the vehicles and having to pay the corresponding penalties. Therefore, neither these accidents are registered nor an exhaustive analysis of the cause of the accident is not carried out by the authorities. In addition, most of these vehicles are commercial vehicles (for freight or passenger transport), so companies carry out their own preventive maintenance programmes to keep them in good condition and not lose working hours that cause economic losses to the company.

The mechanical failure drop in the time series with the quadratic trend in VTIs from 2013 onwards is related to the entry into force of the Resolution No 006-CNC-2012 in April 2012, by means of in which the National Competence Council transferred the competence to plan, regulate and control traffic, land transport and road safety to the municipal decentralised autonomous governments of Ecuador, where there were stricter regulations on traffic conditions and technical condition requirements of vehicles, consequently, the users gave importance to the maintenance of their vehicles, to avoid the corresponding sanctions Furthermore, a decrease in road crashes caused by mechanical failure was observed in all three subcategories analysed.

Regarding braking system failure as the cause of road crashes in subcategory M1, it can be disaggregated into two according to the most common ones recorded in VTIs: insufficient tyre tread and parking brake effectiveness. This mechanical failure matches with what was found in the study by [22], where fatal crashes were caused by tyre failure, which represents two-thirds of the mechanical failures (28%), reinforcing the need for the Vehicle Technical Inspection process.

An important number of studies over the most representative failures came to the conclusion that VTI performance has a positive influence on traffic accidents and the results obtained in this research provide the public and authorities in charge of regulating, controlling traffic and road safety with an insight into the mechanical failure factors that can influence traffic accidents. Therefore, it can be stated that the use of VTI as a mitigating factor in traffic accidents depends on the country and the conditions where it is performed, being recommendable further research to establish a clear correlation between VTI and traffic accidents whenever those databases are available. In developing countries such as Ecuador, it is a limitation to be able to acquire organised and systematised data that is reliable from a single source. For the present research, three entities were used that had information of interest in paper and digital archives that had to be systematised and organised.

## Figures and Tables

**Figure 1 ijerph-19-07787-f001:**
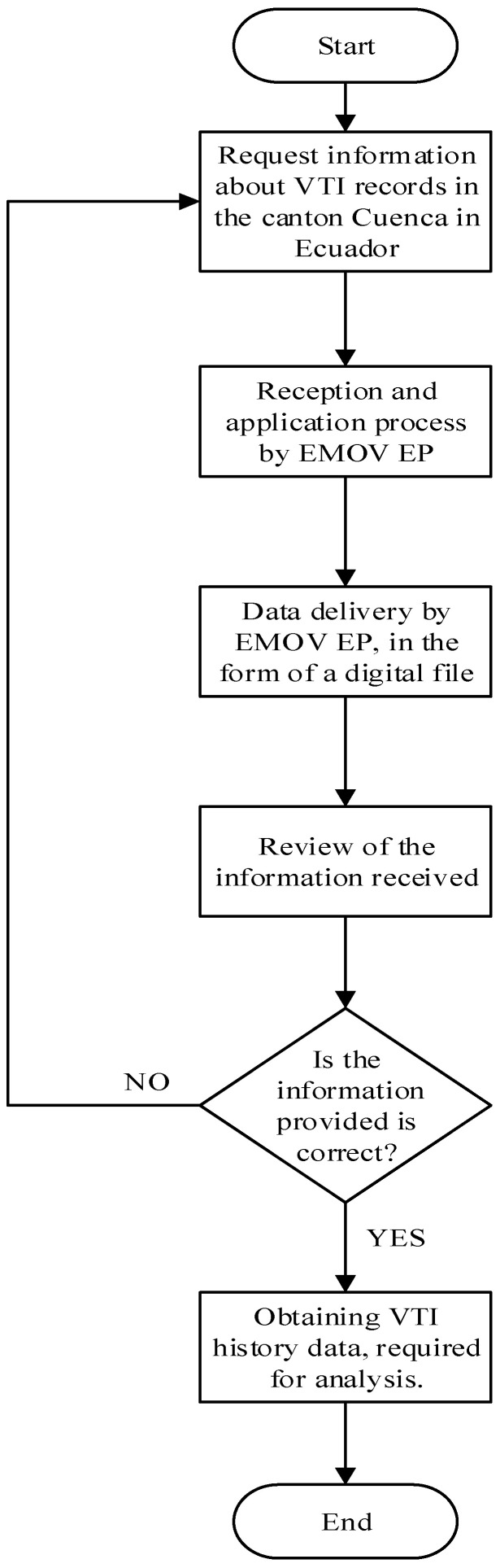
Process for VTI records data collection.

**Figure 2 ijerph-19-07787-f002:**
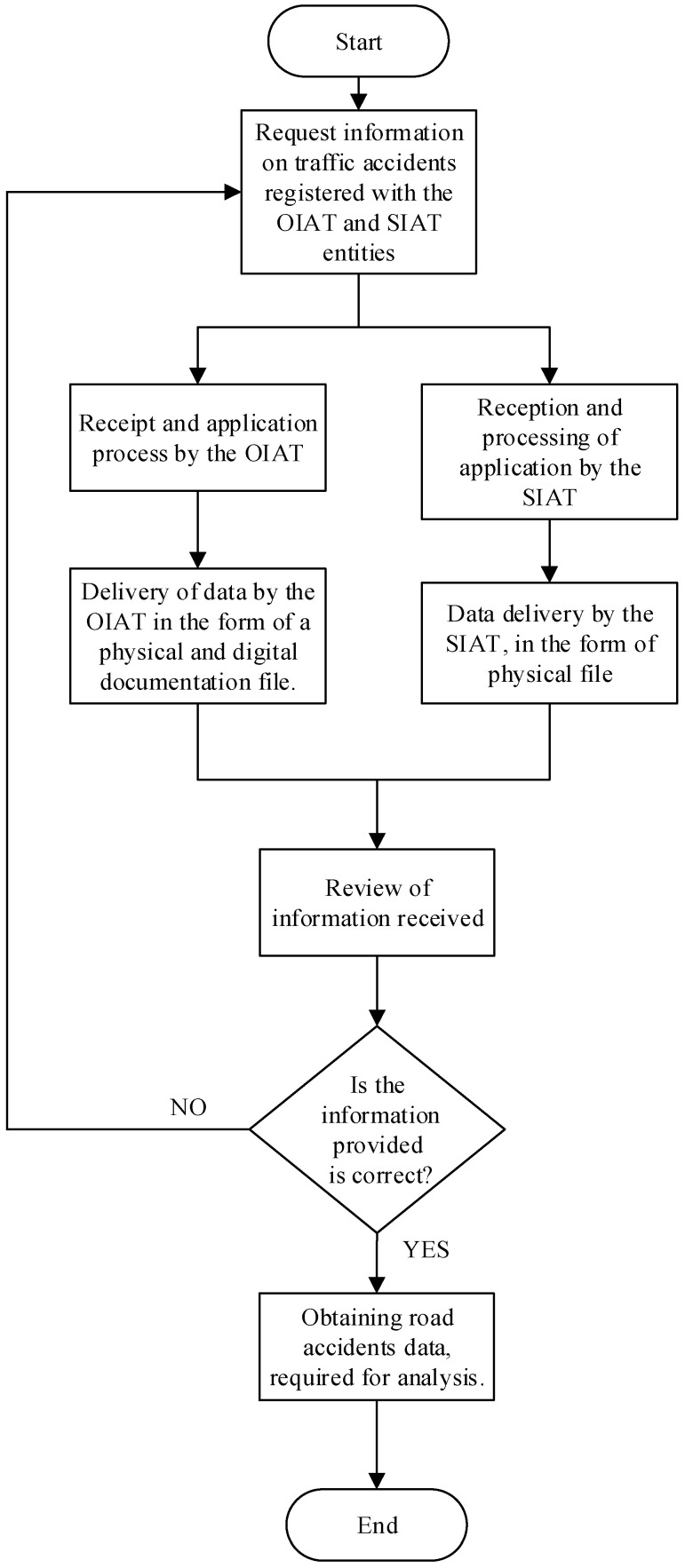
The process to obtain the data regarding road crashes.

**Figure 3 ijerph-19-07787-f003:**
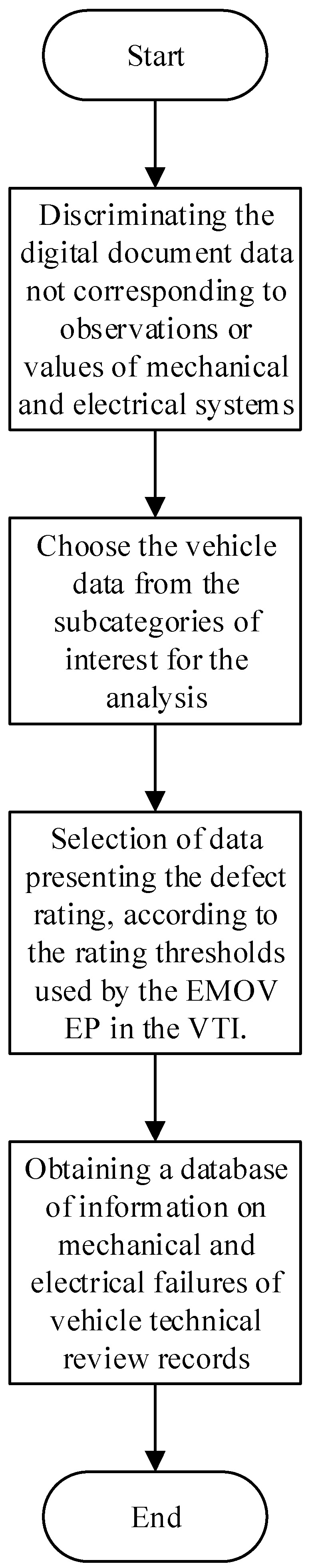
Process for the database creation of the mechanical and electrical failures from the VTI records.

**Figure 4 ijerph-19-07787-f004:**
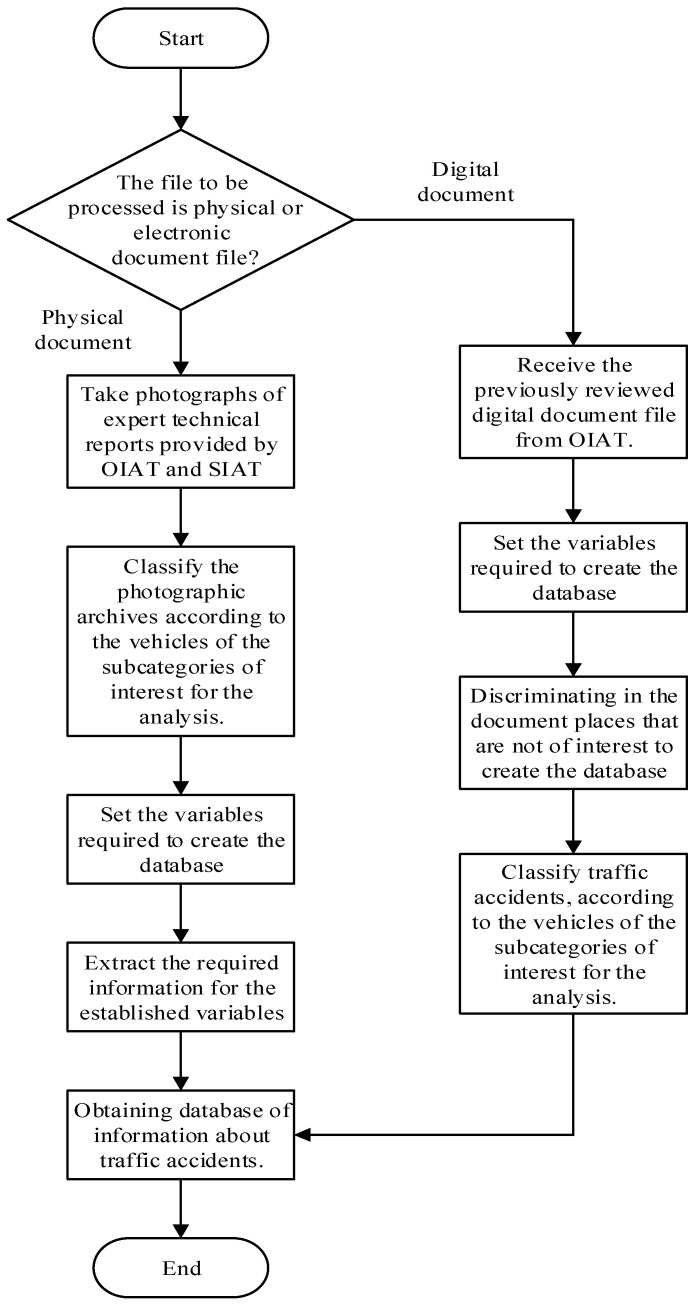
Process for the traffic accident database creation.

**Figure 5 ijerph-19-07787-f005:**
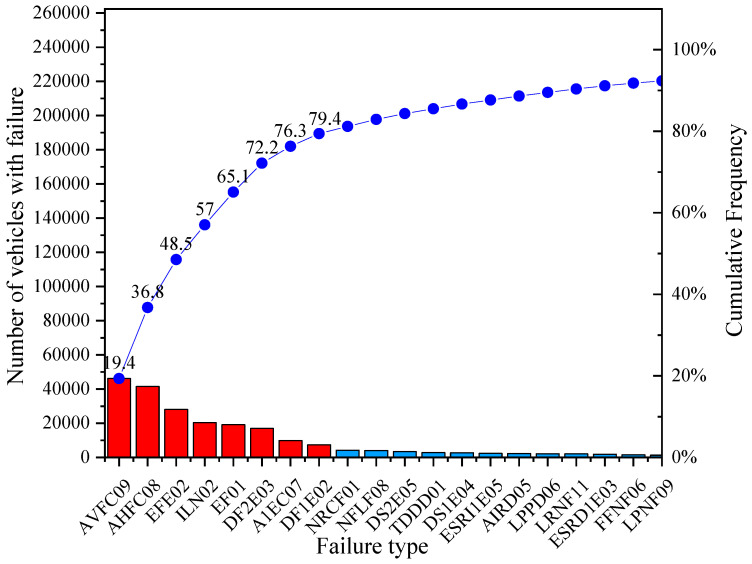
Representative failures in VTI for subcategory M1.

**Figure 6 ijerph-19-07787-f006:**
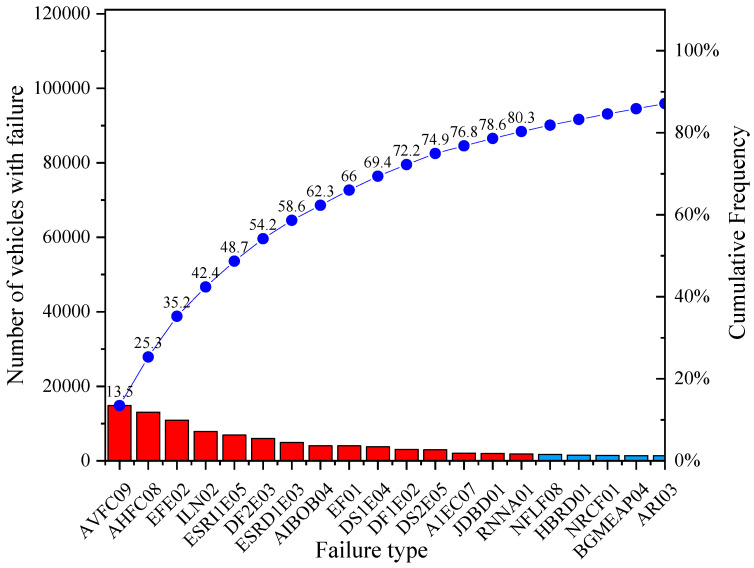
VTI representative failures for subcategory N1.

**Figure 7 ijerph-19-07787-f007:**
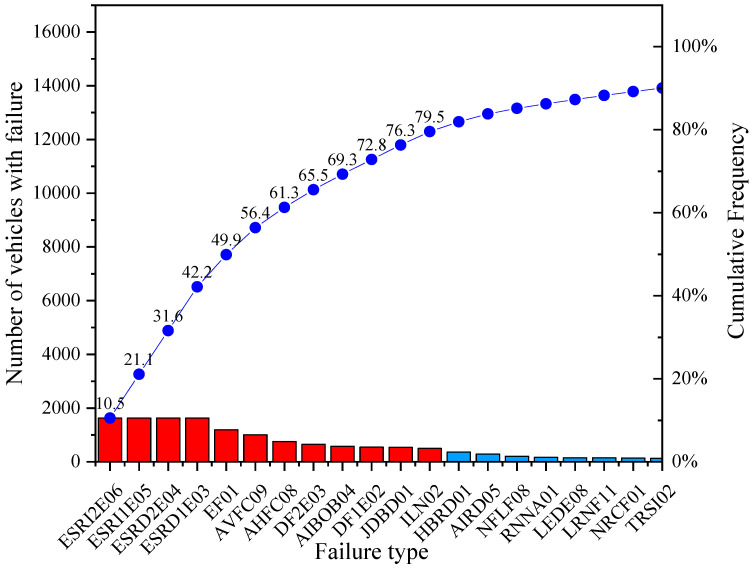
VTI representative failures for subcategory M3 bus type.

**Figure 8 ijerph-19-07787-f008:**
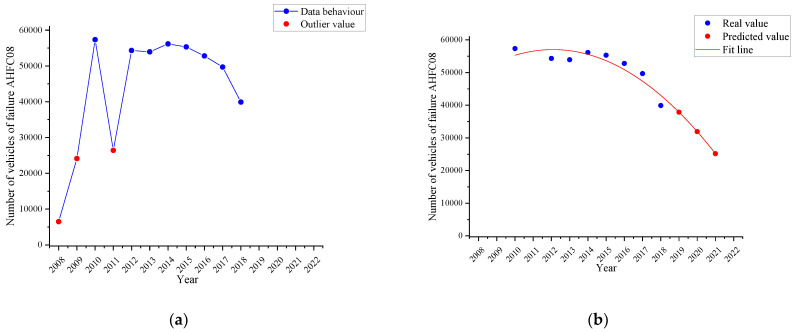
Failure trend named as horizontal alignment of the driver headlight (AHFC08). (**a**): behaviour of the failure AHFC08 (blue line); (**b**): the time series with a quadratic trend (red line) and the forecast (red dots).

**Figure 9 ijerph-19-07787-f009:**
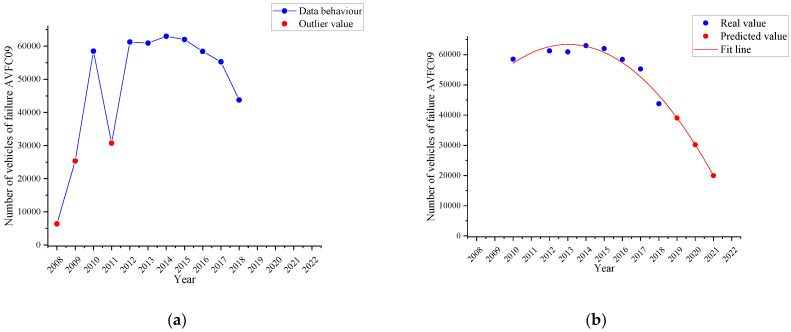
Failure trend named as vertical alignment of the driver headlight (AVFC09). (**a**): behaviour of the failure (blue line); (**b**): the time series with a quadratic trend (red line) and the forecast (red dots).

**Figure 10 ijerph-19-07787-f010:**
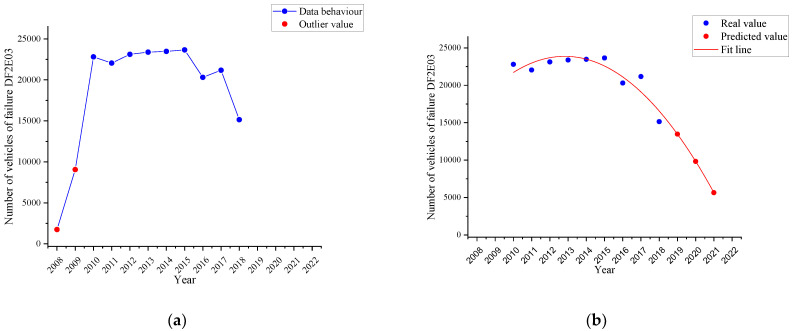
Failure trend is named as a braking imbalance on the 2nd axle (DF2A03). (**a**): Behaviour of the failure (blue line); (**b**): the time series with a quadratic trend (red line) and the forecast (red dots).

**Figure 11 ijerph-19-07787-f011:**
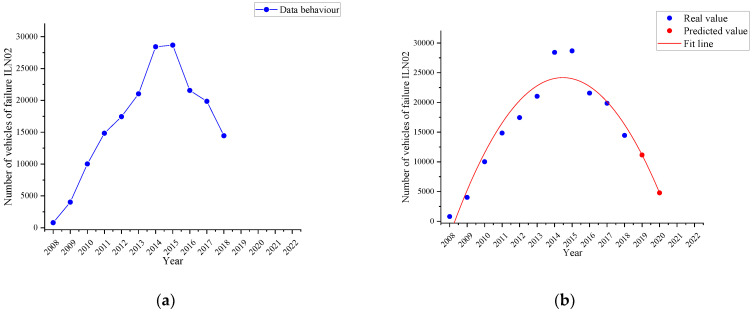
Failure trend named as insufficient tyre tread (ILN02). (**a**): Behaviour of the failure (blue line); (**b**): the time series with a quadratic trend (red line) and the forecast (red dots).

**Figure 12 ijerph-19-07787-f012:**
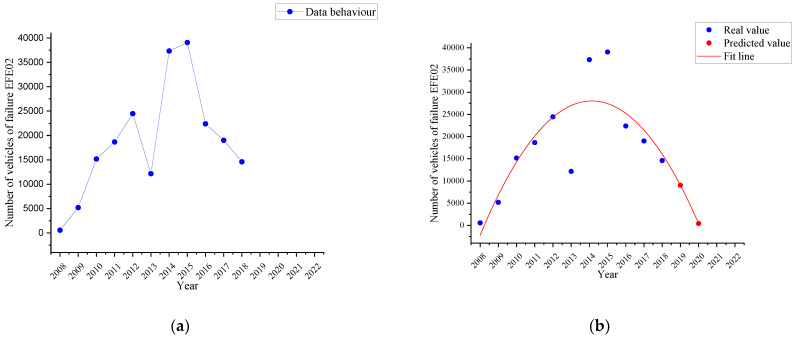
Failure trend named as parking break effectiveness (EFE02). (**a**): Behaviour of the failure (blue line); (**b**): the time series with a quadratic trend (red line) and the forecast (red dots).

**Figure 13 ijerph-19-07787-f013:**
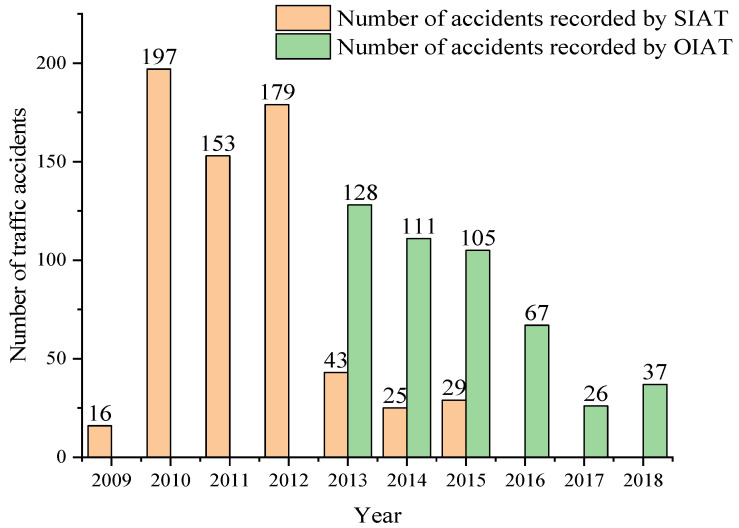
Number of traffic accidents per year for subcategory M1.

**Figure 14 ijerph-19-07787-f014:**
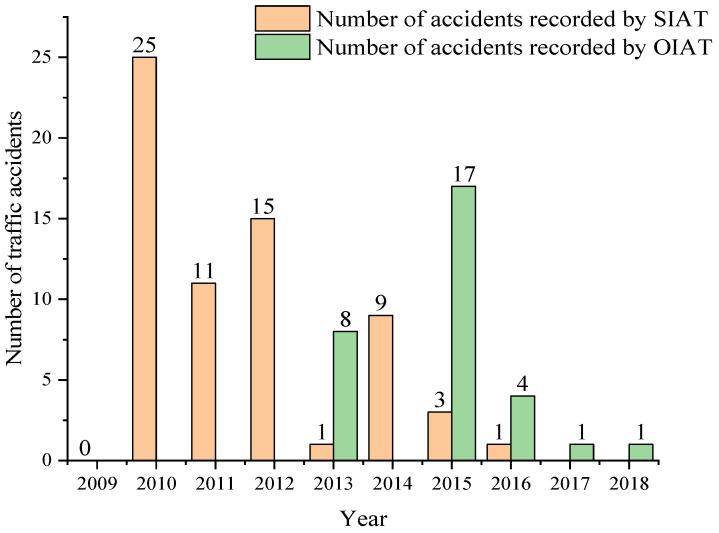
Number of traffic accidents per year for subcategory M3 bus type.

**Figure 15 ijerph-19-07787-f015:**
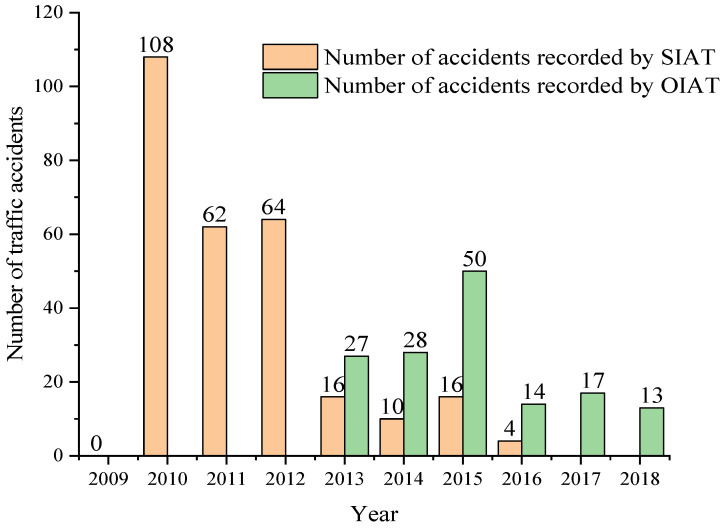
Number of traffic accidents per year for subcategory N1.

**Figure 16 ijerph-19-07787-f016:**
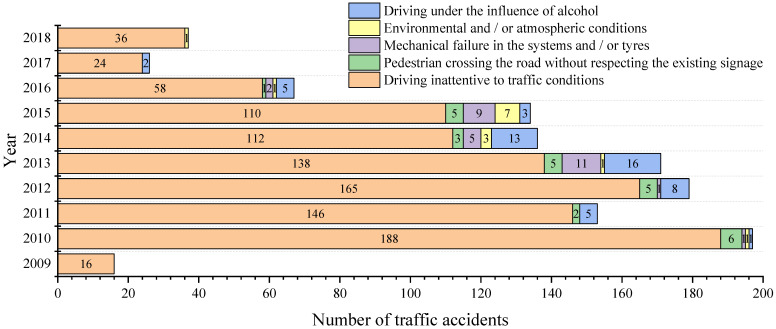
Traffic accident causes per year for subcategory M1.

**Figure 17 ijerph-19-07787-f017:**
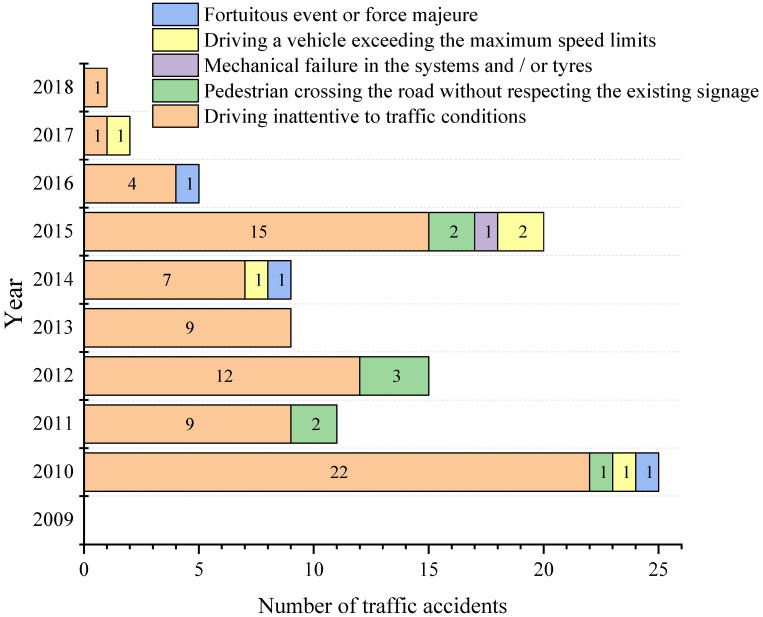
Traffic accident causes per year for subcategory M3 bus type.

**Figure 18 ijerph-19-07787-f018:**
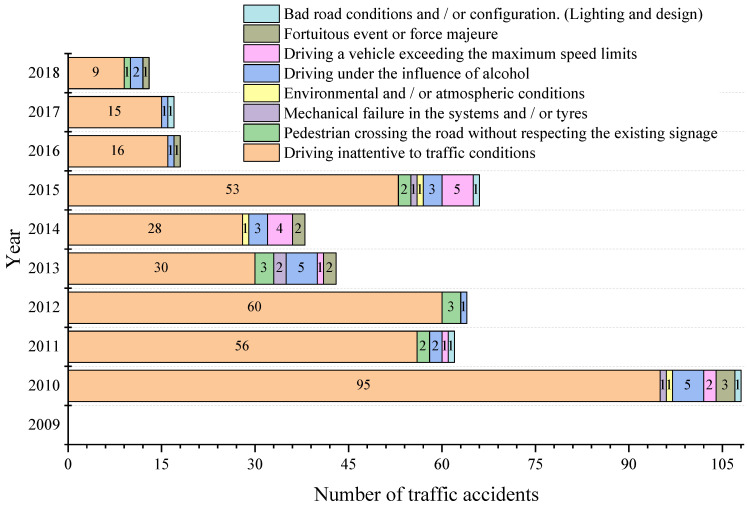
Traffic accident causes per year for subcategory N1.

**Figure 19 ijerph-19-07787-f019:**
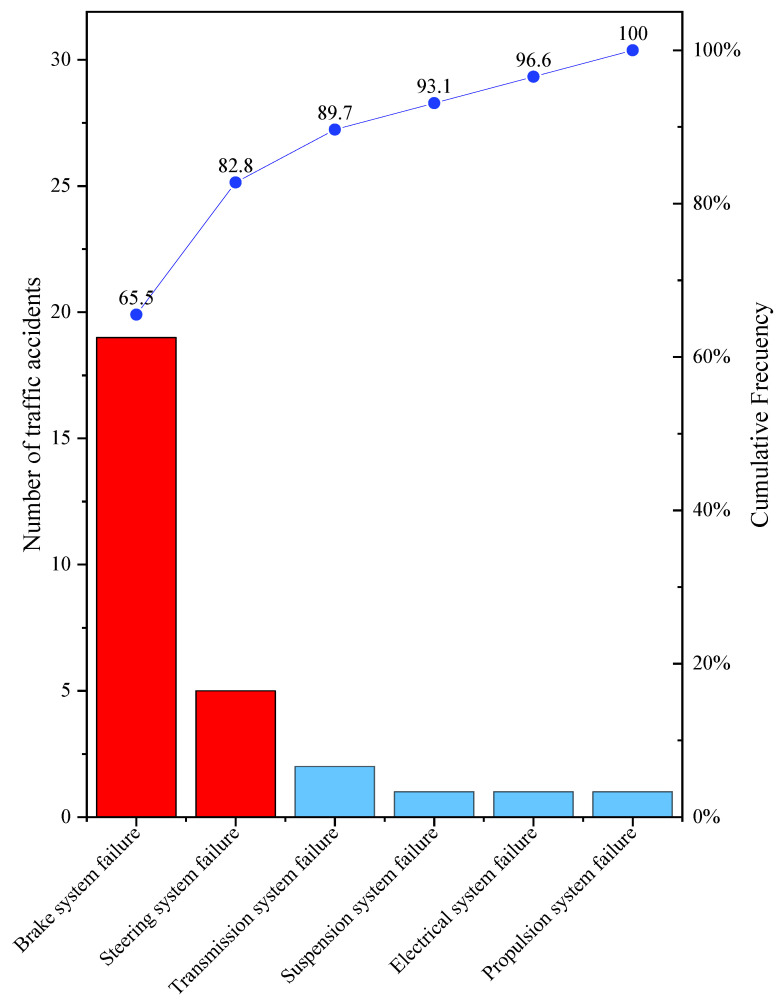
Traffic accident causes recorded for subcategory M1.

**Table 1 ijerph-19-07787-t001:** Mode results.

FAILURE	NUMBER OF VEHICLES
Horizontal alignment of the driver headlight	55,305
Vertical alignment of the driver headlight	62,001
Braking imbalance on the 2nd axle	23,653
Insufficient tyre tread	28,674
Parking brake effectiveness	39,065

## Data Availability

Data sharing not applicable.

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
