# Peer review of "Identification of the Mechanical Failure Factors with Potential Influencing Road Accidents in Ecuador"

_ijerph, 2022, doi:10.3390/ijerph19137787_

Round 1

Reviewer 1 Report

1.     The manuscript presents identification of the mechanical failure factors with potential influencing road accidents in Ecuador, which is interesting.

2.     However, the manuscript, in its present form, contains several weaknesses. Appropriate revisions to the following points should be undertaken in order to justify recommendation for publication.

3.     Full names should be shown for all abbreviations in their first occurrence in texts. For example, SDO in p.3, etc.

4.     For readers to quickly catch your contribution, it would be better to highlight major difficulties and challenges, and your original achievements to overcome them, in a clearer way in abstract and introduction.

5.     It is shown in the reference list that the authors have several publications in this field. This raises some concerns regarding the potential overlap with their previous works. The authors should explicitly state the novel contribution of this work, the similarities, and the differences of this work with their previous publications.

6.     p.1 - the vehicle subcategories M1 and M3 and N1 are adopted in this study. What are the other feasible alternatives? What are the advantages of adopting these subcategories over others in this case? How will this affect the results? More details should be furnished.

7.     p.1 - the city of Cuenca is adopted as the case study. What are other feasible alternatives? What are the advantages of adopting this case study over others in this case? How will this affect the results? The authors should provide more details on this.

8.     p.5 - a database provided by several entities is adopted in the experiments. What are the other feasible alternatives? What are the advantages of adopting this database over others in this case? How will this affect the results? More details should be furnished.

9.     p.5 - historical records of 2008 to 2018 are taken. Why are more recent data not included in the study? Is there any difficulty in obtaining more recent data? Are there any changes to the situation in recent years? What are its effects on the result?

10.  p.10 - the Pareto chart is adopted to identify the most relevant failures of the vehicles in selected subcategories. What are the other feasible alternatives? What are the advantages of adopting this approach over others in this case? How will this affect the results? More details should be furnished.

11.  p.10 - time series analysis with quadratic trend is adopted to fit with the most representative failures recorded in the VTI. What are the other feasible alternatives? What are the advantages of adopting this soft computing technique over others in this case? How will this affect the results? More details should be furnished.

12.  p.13-14 - “…it is suspected that in 2011 there were problems such as… the equipment. For these reasons.…” More justification should be furnished on this issue.

13.  p.15 - the time series with quadratic trend (red line) is adopted to determine the fitting curve. What are the other feasible alternatives? What are the advantages of adopting this function over others in this case? How will this affect the results? More details should be furnished.

14.  p.22 - “…Considering this circumstance, the high rate of this failure may be due to the.…” More justification should be furnished on this issue.

15.  p.23 - “…In both cases, negative values are obtained when calculating an estimate for the year 2021, since the fitting curve shows a very accentuated decreasing gradient. An explanation to this fact may be that.…” More justification should be furnished on this issue.

16.  p.23 - “…For the two remaining subcategories (M3 bus type and N1), there were few accidents due to mechanical failures making difficult to obtain any relevant conclusion. An explanation to this fact is that.…” More justification should be furnished on this issue.

17.  Some key model parameters are not mentioned. The rationale on the choice of the set of parameters should be explained with more details. Have the authors experimented with other sets of values? What are the sensitivities of these parameters on the results?

18.  Some assumptions are stated in various sections. Justifications should be provided on these assumptions. Evaluation on how they will affect the results should be made.

19.  The discussion section in the present form is relatively weak and should be strengthened with more details and justifications.

20.  Moreover, the manuscript could be substantially improved by relying and citing more recent literature about contemporary real-life case studies of soft computing techniques on sustainability and/or uncertainty such as the following. Discussions about result comparison and/or incorporation of those concepts in your works are encouraged:

          Sharafati, A., et al., “A strategy to assess the uncertainty of a climate change impact on extreme hydrological events in the semi-arid Dehbar catchment in Iran,” Theoretical and Applied Climatology 139 (1-2): 389-402 2020.

          Shamshirband, S., et al., “Ensemble models with uncertainty analysis for multi-day ahead forecasting of chlorophyll a concentration in coastal waters,” Engineering Applications of Computational Fluid Mechanics 13 (1): 91-101 2019.

          Aryafar, A., et al., “Evolving genetic programming and other AI-based models for estimating groundwater quality parameters of the Khezri plain, Eastern Iran,” Environmental Earth Sciences 78: article no. 69 2019.

21.  Some inconsistencies and minor errors that needed attention are:

          Replace “…Figure 1, shows the…” with “…Figure 1 shows the…” in line 246 of p.5

          Replace “…In figure 14, it can…” with “…In Figure 14, it can…” in line 501 of p.16

          Replace “…[23], recommends that…” with “…[23] recommends that…” in line 580 of p.21

          and more…

22.  In the conclusion section, the limitations of this study and suggested improvements of this work should be highlighted.

Author Response

Response to Reviewer 1 Comments

  1. The manuscript presents identification of the mechanical failure factors with potential influencing road accidents in Ecuador, which is interesting.

Response 1: Thank you very much for your comment.

  1. However, the manuscript, in its present form, contains several weaknesses. Appropriate revisions to the following points should be undertaken in order to justify recommendation for publication.

Response 2: Thank you very much for your comments to improve the quality of the article.

  1. Full names should be shown for all abbreviations in their first occurrence in texts. For example, SDO in p.3, etc.

Response 3: Your observation is considered, the abbreviation is deleted and replaced by the full name, as it does not appear in any other paragraph of the document.

  1. For readers to quickly catch your contribution, it would be better to highlight major difficulties and challenges, and your original achievements to overcome them, in a clearer way in abstract and introduction.

Response 4: The observation is considered. In order to clarify the contribution, items highlighting the main difficulties and challenges are added at the beginning of the abstract. Furthermore, in the introduction, paragraphs 6 and 13 are added, thus contributing to resolving the observation raised.

  1. It is shown in the reference list that the authors have several publications in this field. This raises some concerns regarding the potential overlap with their previous works. The authors should explicitly state the novel contribution of this work, the similarities, and the differences of this work with their previous publications.

Response 5: Your observation is considered. In order to clarify the contribution, some items highlighting the contribution of present research are added at the beginning of the abstract. Last paragraph of the conclusion is also rewritten, highlighting the contribution of this research according to the obtained outcomes. This marks a clear difference with previous work in the area of study.

  1. p.1 - the vehicle subcategories M1 and M3 and N1 are adopted in this study. What are the other feasible alternatives? What are the advantages of adopting these subcategories over others in this case? How will this affect the results? More details should be furnished.

Response 6: Another alternative could be subcategory L1 (Bicimoto / Moped), according to the Ecuadorian technical standard NTE INEN 2656.

The reason to adopt M1, M3 and N1 is that these subcategories represent the largest vehicle fleet size (70 % of vehicles) with a total of 59 % of accidents in Ecuador, which is detailed in the introduction section in the paragraphs 10, 11 and 17. “Private vehicles have the highest representation in the countries' vehicle fleet and as vehicles involved in road crashes. The risk of dying in a road crash as private vehicle or bus passengers is 10 times higher”.

If this methodology were applied to subcategory L1, the impact on the results would not be significant. The other subcategories are not addressed because they do not correspond to the purpose of this research.

  1. p.1 - the city of Cuenca is adopted as the case study. What are other feasible alternatives? What are the advantages of adopting this case study over others in this case? How will this affect the results? The authors should provide more details on this.

Response 7: The main reason to adopt the city of Cuenca as the case study was the data availability. Data were available for research thanks to the agreement between the governmental and academic entities of the city, being a matter of interest. There were no other records available with such detailed information as for Cuenca, to be considered in the study. 

In order to clarify this question, we have added at the beginning of section 3.1 a new paragraph to provide more details.

  1. p.5 - a database provided by several entities is adopted in the experiments. What are the other feasible alternatives? What are the advantages of adopting this database over others in this case? How will this affect the results? More details should be furnished.

Response 8: The answer to this question is directly related to the previous one. It has been considered, in first paragraph of section 3.1

  1. p.5 - historical records of 2008 to 2018 are taken. Why are more recent data not included in the study? Is there any difficulty in obtaining more recent data? Are there any changes to the situation in recent years? What are its effects on the result?

Response 9: This is a good point, but difficult to solve by researchers. Update of data base was on delayed when COVID pandemic interrupted normal life. Some activities, such as vehicle accident analysis and registration, move into the background. Nevertheless, this observation is considered in section 3.1. A justification is added in second and third paragraphs in order to answer the questions posed.

  1. p.10 - the Pareto chart is adopted to identify the most relevant failures of the vehicles in selected subcategories. What are the other feasible alternatives? What are the advantages of adopting this approach over others in this case? How will this affect the results? More details should be furnished.

Response 10: Another feasible alternative could be the histogram.  However, after the literature review, it was found that this was not the most appropriate for this study, as it does not represent the results as well as the Pareto diagram. As set up in the first paragraph of section 3.3.2, Pareto chart has the advantage of being an important tool for its ability to focus attention on the area of interest, separating the significant factors and identifying the most important sources of problems. The use of Pareto chart in previous researches in the area of transport demonstrates the usefulness of this simple tool to identify the most relevant parameters within a big sample size. 

  1. p.10 - time series analysis with quadratic trend is adopted to fit with the most representative failures recorded in the VTI. What are the other feasible alternatives? What are the advantages of adopting this soft computing technique over others in this case? How will this affect the results? More details should be furnished.

Response 11: We understand that, effectively, there are other alternatives and computational prediction models such as neural networks (ANN), that probably also fit for the same purpose.  However, considering the requirements of research in reference to the behavior of the data, it was dismissed due to complexity. Time series analysis with quadratic trend has the advantage of simplicity. As mentioned in the first paragraph of section 3.3.3, it is possible to analyse and model an ordered sequence of observations over time, allowing the understanding and modelling of the data generation system and the forecasting of future values in a clear and simple way compared to the application of neural networks. In addition, several research found in literature show the usefulness of this tool in the area of transport.

Maybe, a question to be considered for further research should be the benefits to apply neural networks to predict vehicle crashes behavior over time.

  1. p.13-14 - “…it is suspected that in 2011 there were problems such as… the equipment. For these reasons.…” More justification should be furnished on this issue.

Response 12: the observation is considered; the first paragraph of section 4.2.1 is rewritten to justify the issue.

  1. p.15 - the time series with quadratic trend (red line) is adopted to determine the fitting curve. What are the other feasible alternatives? What are the advantages of adopting this function over others in this case? How will this affect the results? More details should be furnished.

Response 13: It is not feasible to use other trends to determine the behaviour of the data, as values with another trend are outside the actual estimates. As it is indicated in the second paragraph of section 4.2, a good data fit for each one of the models was obtained with an R2 greater than 0.8, generating reliable forecasts.

  1. p.22 - “…Considering this circumstance, the high rate of this failure may be due to the.…” More justification should be furnished on this issue.

Response 14: Thanks for the observation. It has been considered in the text. Paragraph 2 of the conclusions is rewritten with the aim of justifying the conclusion

  1. p.23 - “…In both cases, negative values are obtained when calculating an estimate for the year 2021, since the fitting curve shows a very accentuated decreasing gradient. An explanation to this fact may be that.…” More justification should be furnished on this issue.

Response 15: Thanks for the observation. It has been considered in the text. Paragraph 5 of the conclusions is rewritten to better justify this issue.

  1. p.23 - “…For the two remaining subcategories (M3 bus type and N1), there were few accidents due to mechanical failures making difficult to obtain any relevant conclusion. An explanation to this fact is that.…” More justification should be furnished on this issue.

Response 16: Thanks for the observation. It has been considered in the text. Paragraph 8 of the conclusions is rewritten to better justify this issue.

  1. Some key model parameters are not mentioned. The rationale on the choice of the set of parameters should be explained with more details. Have the authors experimented with other sets of values? What are the sensitivities of these parameters on the results?

Response 17: Thanks again for the observation. A new sentence is added at the end of the first paragraph of section 4.1 to clarify the choice of parameters and a translation error is corrected in this paragraph. Regarding the use of other sets of values, there is no need to experiment with other results as the most representative ones were chosen by applying the mode, and there is no need to apply sensitivity analysis to this data set for this research.

  1. Some assumptions are stated in various sections. Justifications should be provided on these assumptions. Evaluation on how they will affect the results should be made.

Response 18: Your observation is considered. Paragraphs 3 and 9 in the conclusions, where assumptions are made were corrected and do not affect the results.

  1. The discussion section in the present form is relatively weak and should be strengthened with more details and justifications.

Response 19: In this section, the results are discussed taking into account the question stated at the beginning of this research, regarding the ability of the vehicle revision system to detect technical failures and subsequently proceed to correct these failures. The results and their possible interpretation are posed from the perspective of other existing studies in the same area and considering additional influential parameters related to the state of art and technic of the location where the VTI is implemented. However and also related to suggestion 20, in order to reinforce this section a last paragraph is added, posing the need for further research using other techniques and models of analysis, setting up the direction of future research.

  1. Moreover, the manuscript could be substantially improved by relying and citing more recent literature about contemporary real-life case studies of soft computing techniques on sustainability and/or uncertainty such as the following. Discussions about result comparison and/or incorporation of those concepts in your works are encouraged:
  • Sharafati, A., et al., “A strategy to assess the uncertainty of a climate change impact on extreme hydrological events in the semi-arid Dehbar catchment in Iran,” Theoretical and Applied Climatology 139 (1-2): 389-402 2020.
  • Shamshirband, S., et al., “Ensemble models with uncertainty analysis for multi-day ahead forecasting of chlorophyll a concentration in coastal waters,” Engineering Applications of Computational Fluid Mechanics 13 (1): 91-101 2019.
  • Aryafar, A., et al., “Evolving genetic programming and other AI-based models for estimating groundwater quality parameters of the Khezri plain, Eastern Iran,” Environmental Earth Sciences 78: article no. 69 2019.

Response 20: Thanks for the observation. It has been considered in the text. The last paragraph is added to the discussion section, to incorporate the use of other techniques of analysis of the suggested studies in future research to follow up this work.

  1. Some inconsistencies and minor errors that needed attention are:
  • Replace “…Figure 1, shows the…” with “…Figure 1 shows the…” in line 246 of p.5

Response 21.1: the observation is considered; correction was made.

  • Replace “…In figure 14, it can…” with “…In Figure 14, it can…” in line 501 of p.16

Response 21.2: the observation is considered; correction was made.

  • Replace “…[23], recommends that…” with “…[23] recommends that…” in line 580 of p.21

Response 21.3: the observation is considered; correction was made.

  • and more…

Response 21.4: the observation is considered, on the lines where inconsistencies and minor errors are found, these are corrected.

  1. In the conclusion section, the limitations of this study and suggested improvements of this work should be highlighted.

Response 22: Thank you for your comment. It has been taken into account in the text. Last paragraph of conclusions is rewritten, highlighting future improvements for the research and limitations encountered.

Reviewer 2 Report

Dear Author(s),

I would like to thank you for the opportunity to read your manuscript entitled “Identification of the mechanical failure factors with potential influencing road accidents in Ecuador”.

The overall manuscript is well presented with minor spelling or grammar mistakes.

The overall work is very interesting, as the problem of road safety is very relevant and important.  

Here are some issues concerning your paper:

1.      The overall purpose of the article is stated in the abstract as “to setup the procedure to identify the mechanical failures that contribute to traffic accidents in cities located in developing countries, with the city of Cuenca - Ecuador” (lines ) but in the introduction it is different “the objective of this research is to identify the most common failures that cause or may cause a traffic accident and try to give an answer to this question in cities located in developing countries, in the context of the vehicle fleet in the city of Cuenca - Ecuador” (lines 131-133). It is confusing as the research aims stated in abstract and introduction are similar but not the same.

2.      The Literature Review part is logical and well organized, but some elements are missing. What is the gap knowledge of publications cited? Is there a difference of the methodological approach?

3.      Change “si” into “yes” in the chart presented in Figure 1.

4.      According to the charts in Figures 5-7: what is on the OY axis? Failure type probably? It is not clear for the reader. I could only guess it because of next figures.

5.      Text references to charts in Figure 14 (line 501) and Figure 15 (line 504) should be capitalized.

6.      Future research directions and the significance of the results of the research achieved should be underlined and explained in conclusion part.

Author Response

Response to Reviewer 2 Comments

  1. The overall purpose of the article is stated in the abstract as “to setup the procedure to identify the mechanical failures that contribute to traffic accidents in cities located in developing countries, with the city of Cuenca - Ecuador” (lines ) but in the introduction it is different “the objective of this research is to identify the most common failures that cause or may cause a traffic accident and try to give an answer to this question in cities located in developing countries, in the context of the vehicle fleet in the city of Cuenca - Ecuador” (lines 131-133). It is confusing as the research aims stated in abstract and introduction are similar but not the same.

Response 1: Thanks for your observation. It has been considered in text. 16th paragraph in the introduction was rewritten to match the aim of the abstract.

  1. The Literature Review part is logical and well organized, but some elements are missing. What is the gap knowledge of publications cited? Is there a difference of the methodological approach?

Response 2: Thanks for the observation. It has been considered in the text. Items highlighting the main difficulties and challenges are added at the beginning of the abstract, paragraphs 6 and 13 are added in the introduction to clarify the contribution of the research to avoid possible gaps between the cited publications and about the methodological approach.

  1. Change “si” into “yes” in the chart presented in Figure 1.

Response 3: Thanks for your observation. Figure 1 was corrected.

  1. According to the charts in Figures 5-7: what is on the OY axis? Failure type probably? It is not clear for the reader. I could only guess it because of next figures.

Response 4: Thanks for your observation. Figures 5-7 were corrected.

  1. Text references to charts in Figure 14 (line 501) and Figure 15 (line 504) should be capitalized.

Response 5: Thanks again for correction. Textual references to graphics in Figure 14 and Figure 15 are changed to capitalized.

  1. Future research directions and the significance of the results of the research achieved should be underlined and explained in conclusion part.

Response 6: It has been also done. Last paragraph of conclusions is rewritten, underlining the results obtained and setting out future research on the subject.

Round 2

Reviewer 1 Report

The revised paper has addressed all my previous comments, and I suggest to ACCEPT the paper as it is now.